# LncRNA *SMARCD3-OT1* Promotes Muscle Hypertrophy and Fast-Twitch Fiber Transformation via Enhancing *SMARCD3X4* Expression

**DOI:** 10.3390/ijms23094510

**Published:** 2022-04-19

**Authors:** Jing Zhang, Bolin Cai, Manting Ma, Shaofen Kong, Zhen Zhou, Xiquan Zhang, Qinghua Nie

**Affiliations:** 1Department of Animal Genetics, Breeding and Reproduction, College of Animal Science, South China Agricultural University, Guangzhou 510642, China; jane_zhang233@hotmail.com (J.Z.); bolincai@scau.edu.cn (B.C.); mamanting@stu.scau.edu.cn (M.M.); shaofenkong@163.com (S.K.); zhenzhou@stu.scau.edu.cn (Z.Z.); xqzhang@scau.edu.cn (X.Z.); 2Guangdong Provincial Key Lab of Agro-Animal Genomics and Molecular Breeding, and Key Laboratory of Chicken Genetics, Breeding and Reproduction, Ministry of Agriculture, Guangzhou 510642, China; 3National-Local Joint Engineering Research Center for Livestock Breeding, College of Animal Science, South China Agricultural University, Guangzhou 510642, China

**Keywords:** lncRNA, myogenesis, myoblast, cell proliferation, cell differentiation, *CDKN1A*, *MYOD*, *SMARCD3*

## Abstract

Long noncoding RNA (lncRNA) plays a crucial part in all kinds of life activities, especially in myogenesis. *SMARCD3* (SWI/SNF-related, matrix-associated, actin-dependent regulator of chromatin, subfamily d, member 3) is a member of the SWI/SNF protein complex and was reported to be required for cell proliferation and myoblast differentiation. In this study, we identified a new lncRNA named *SMARCD3-OT1* (*SMARCD3*
*overlapping*
*lncRNA*), which strongly regulated the development of myogenesis by improving the expression of *SMARCD3X4* (*SMARCD3*
*transcripts*
*4*). We overexpressed and knockdown the expression of *SMARCD3-OT1* and *SMARCD3X4* to investigate their function on myoblast proliferation and differentiation. Cell experiments proved that *SMARCD3-OT1* and *SMARCD3X4* promoted myoblast proliferation through the *CDKN1A* pathway and improved differentiation of differentiated myoblasts through the *MYOD* pathway. Moreover, they upregulated the fast-twitch fiber-related genes and downregulated the slow-twitch fiber-related genes, which indicated that they facilitated the slow-twitch fiber to transform into the fast-twitch fiber. The animals’ experiments supported the results above, demonstrating that *SMARCD3-OT1* could induce muscle hypertrophy and fast-twitch fiber transformation. In conclusion, *SMARCD3-OT1* can improve the expression of *SMARCD3X4*, thus inducing muscle hypertrophy. In addition, *SMARCD3-OT1* can facilitate slow-twitch fibers to transform into fast-twitch fibers.

## 1. Introduction

Myogenesis is a highly ordered process. A series of myogenic factors regulates basic activities inside the muscle. The number of skeletal muscles is almost fixed in utero, and the key to muscle weight gain is the size of myofibers, which can be changed with exercise [1,2]. In addition, skeletal muscle is dynamic muscle tissue. Under certain conditions, fast-twitch fibers and slow-twitch fibers can be transformed into each other. Different types of muscle fibers have different functional, biochemical, and morphological characteristics [3].

Although lncRNAs have similar parts with mRNA, lncRNAs have more flexible regulation methods and more significant differences in expression between tissues [4,5,6]. These characteristics make the study of lncRNA more difficult and complex. Recently, more and more lncRNAs were found to be relevant to epigenetic and transcriptional regulation in myogenesis [7,8,9,10]. Studies have shown that some lncRNAs could activate the expression of adjacent encoding genes to perform different functions on life activities [11,12].

The protein encoded by the *SMARCD3* gene is a member of the SWI/SNF protein family. Its members have helicase and ATPase activities and can regulate the transcription of genes by changing the chromatin structure around the genes, playing an important role in physiological processes [13,14,15]. Reports show that *SMARCD3* can regulate the cell cycle through the *CDKN1A* signaling pathway, affecting tumor formation [16], and it can also regulate muscle development in zebrafish through *MYOD* and *MYF5* signaling pathways [17,18]. However, the specific molecular mechanism of *SMARCD3* is still unknown.

RNA-sequencing based on the pectoral muscle and soleus of Xinghua chicken was performed (ID 751251-BioProject-NCBI (nih.gov)/accession number PRJNA751251/accessed date: 1 September 2021) in the previous experiment. Based on these results, a new lncRNA, *MSTRG.2872.11*, partly overlays on a functional gene named *SMARCD3X4* and is identified and named *SMARCD3-OT1*. *SMARCD3-OT1* is found highly expressed in skeletal muscle and upregulated in myoblast differentiation. In vitro, overexpression and knockdown experiments show that *SMARCD3-OT1* promotes myoblast proliferation during the proliferation stage and improves myoblast differentiation during the differentiation stage. In vivo, *SMARCD3-OT1* can facilitate slow-twitch fiber transfer to fast-twitch fiber and induce muscle hypertrophy. The research reveals that *SMARCD3-OT1* promotes the expression of *SMARCD3X4* in both mRNA and protein levels, improving myoblast proliferation and myotube differentiation through the *CDKN1A* pathway and *MYOD* pathway, respectively. Overall, our studies identify a novel lncRNA, which regulates myogenesis in myoblasts through post-transcriptional gene regulation and affects the transformation of myofibers, providing us a new therapeutic target of muscle atrophy therapy.

## 2. Results

### 2.1. Identification and Characterization of SMARCD3-OT1

For exploring the function of *SMARCD3-OT1*, the 5′ and 3′ ends of *SMARCD3-OT1* were amplificated by the rapid amplification of cDNA ends (RACE) assay (Figure 1a). The sequence of *SMARCD3-OT1* is offered in Appendix A. The National Center for Biotechnology Information was used to confirm the position of *SMARCD3-OT1*. The results showed that *SMARCD3-OT1* consisted of 1542 bases and was located at chicken chromosome 2 (chr2: 118054–118633, 133484–133684, 134874–135641). The results also revealed that *SMARCD3-OT1* partly overlaps on the 5′ untranslated region (UTR) of the mRNA *SMARCD3X4* (Figure 1b). A prediction was performed in Coding Potential Calculator [19] (Figure 1c) to predict the coding ability of lncRNA *SMARCD3-OT1*. Moreover, the termination codon of the ORFs of *SMARCD3-OT1* and the initiation codon of the EGFP gene were mutated to construct a series of ORFs-EGFP fusion protein vectors, which are subsequently transfected into chicken’s primary myoblasts (CPMs). The sequences of ORFs are listed in Appendix A. The proteins acquired from those cells were used for Western blot, which proved that *SMARCD3-OT1* did not have coding ability (Figure 1d,e).

In the previous RNA sequencing (ID 751251-BioProject-NCBI (nih.gov)/accession number PRJNA751251/accessed date: 1 September 2021), *SMARCD3-OT1* was found highly expressed in pectoral muscle, and the subsequent quantitative real-time polymerase chain reaction (qRT-PCR) results proved this tendency (Figure 1f,g). The expression of *SMARCD3-OT1* was found to keep a steady level in the proliferation stage (CPMs cultured in growth medium (GM)) and was significantly upregulated during myoblast differentiation (CPMs cultured in differentiation medium (DM)) (Figure 1h). Otherwise, *SMARCD3-OT1* was highly expressed in breast muscle and leg muscle, indicating that they may participate in the myogenesis of skeletal muscle (Figure 1i). Meanwhile, the expression levels of *SMARCD3-OT1* increased from embryonic day 10 (E10) to E15 (Figure 1j). Cell-fractionation assays suggested that *SMARCD3-OT1* mainly existed in the nuclei of CPMs (Figure 1k,l).

### 2.2. SMARCD3-OT1 Promotes the Proliferation of Myoblasts

CPMs were transfected with the overexpression vector or the antisense oligonucleotides (ASO) fragment of *SMARCD3-OT1* when the CPMs achieved 80–90% confluence. After 48 h, the cells were collected and used for subsequent experiments. The fold-changes of *SMARCD3-OT1* were measured by qRT-PCR (Appendix A). The results obtained from the qRT-PCR are presented in that overexpression of *SMARCD3-OT1* significantly downregulated the expression level of *CDKN1A*, an upstream inhibitory gene of the cell cycle. Then, *SMARCD3-OT1* upregulated the cell-cycle related genes (*CCKN1B*, *CCNA1*, *CCNE*, and *CCND1*), improving myoblast proliferation. On the contrary, knockdown of *SMARCD3-OT1* increased *CDKN1A* expression and inhibited other cell-cycle-related genes, repressing the proliferation of myoblasts. (Figure 2a,b). The cell cycle phase assay suggested that *SMARCD3-OT1* prolonged the S phase of cells, preventing cells from entering the G2M phase and promoting the proliferation of cells (Figure 2c,d). In order to further assess the function of *SMARCD3-OT1*, the cell counting kit-8 (CCK-8) assay was performed to measure the proliferation condition of myoblasts after transfection. It can be seen from these data that overexpression of *SMARCD3-OT1* significantly improved cell proliferation after 36 h of transfection, while inhibition of *SMARCD3-OT1* repressed cell proliferation at 48 h (Figure 2e,f). These results indicated that *SMARCD3-OT1* regulated the cell cycle in the early stage of cell proliferation. In addition, the 5-ethynyl-2′-deoxyuridine (EdU) assay was used to verify the results above. Distinct improvement of cell proliferation was observed after overexpression of *SMARCD3-OT1*, and the proportion of proliferating cells was increased under the regulation of *SMARCD3-OT1*. By contrast, the knockdown of *SMARCD3-OT1* had inverse results (Figure 2g–j). These data suggest that *SMARCD3-OT1* improves myoblast proliferation during myoblast proliferation stage (48 h after CPMs achieved 80–90% confluence).

### 2.3. SMARCD3-OT1 Promotes the Differentiation of Myoblasts during the Differentiation Stage, Facilitating the Myofiber Transformation

After 48 h transfection, CPMs were induced to differentiation for 3 days using the differentiated medium. After 3 days of differentiation, the myoblasts fused together, and myotubes were formed. Then, the expression levels of differentiation marker genes, including *MyHC*, *MYOD*, *MYOG*, and *MYF5*, were detected by qRT-PCR and Western blot to assess the function of *SMARCD3-OT1* on differentiation. Overexpression of *SMARCD3-OT1* significantly increased the expression level of these differentiation-related genes in mRNA level, while knockdown of *SMARCD3-OT1* had inverse results (Figure 3a,b). At the protein level, *SMARCD3-OT1* improved these genes’ expression level as well (Figure 3c,d). Moreover, immunofluorescence staining was performed to investigate the function of *SMARCD3-OT1* in differentiation. The results demonstrated that overexpression of *SMARCD3-OT1* could promote the differentiation of myoblasts and the formation of myotubes, and knockdown of *SMARCD3-OT1* had a negative influence on myoblast differentiation and myofiber mature (Figure 3e–h). These data demonstrate that *SMARCD3-OT1* improves myoblast differentiation after 3 days of differentiation induction, which means that *SMARCD3-OT1* has different functions in the myoblast proliferation and differentiation stage.

Otherwise, we also found that *SMARCD3-OT1* improved the expression of fast-twitch fiber-related genes (*TNNT3*, *TNNC2*, and *SRL*) and inhibited the slow-twitch fiber-related genes (*TNNI1*, *TNNT1*, and *TNNC1*) (Figure 3i,j). These results indicate that *SMARCD3-OT1* facilitates slow-twitch fibers to transfer to fast-twitch fibers.

### 2.4. SMARCD3-OT1 Positively Regulated SMARCD3X4 Transcriptional Activity

Because *SMARCD3-OT1* is located in the 5′ UTR of *SMARCD3X4*, we hypothesized that *SMARCD3-OT1* might regulate *SMARCD3X4* by the cisregulating way (Figure 1b). First, we investigated the expression pattern of *SMARCD3X4*. The data revealed that *SMARCD3X4* was highly expressed in early cell differentiation, and *SMARCD3X4* was highly expressed in breast muscle and leg muscle (Appendix A). Otherwise, the expression of *SMARCD3X4* increased from E13 and sharply increased in E15 (Appendix A). RNA of the nucleus and cytoplasm were separated and used to measure the expression of *SMARCD3X4* in the cell. The data revealed that *SMARCD3X4* mainly existed in the nucleus (Appendix A). The results above illustrate a similar expression pattern between *SMARCD3X*-*OT1* and *SMARCD3X4*, which indicates the relationship between SMARCD3X-*OT1* and *SMARCD3X4*.

The qRT-PCR results showed that the mRNA expression level of *SMARCD3X4* was upregulated with the overexpression of *SMARCD3-OT1* and downregulated with the knockdown of *SMARCD3-OT1* in CPMs (Figure 4a,b). The results of the Western blot also proved that the protein level of SMARCD3X4 was positively correlated with *SMARCD3-OT1* in CPMs (Figure 4c,d). In addition, the experiments in vivo proved this tendency as well (Figure 4e–h). To further ensure the functional fragment of *SMARCD3-OT1*, the full-length sequence of *SMARCD3-OT1* was divided into five fragments as follows: E1 (1–768 bp), E2 (769–964 bp), E3 (965–1542 bp), E1 + 2 (1–964 bp), and E1 + 2 + 3 (1–1542 bp). These fragments were inserted into the 5′ UTR of the luciferase gene and then transfected in the DF-1 cell line (Figure 4i). The luciferase assay showed that fragment E3 significantly improved luciferase activity, while fragment E1 + 2 significantly inhibited luciferase activity. The data indicated that fragment E3 might play a positive role in the expression of *SMARCD3X4*, while fragment E1 + 2 has a negative influence on the expression of *SMARCD3X4*. Fragment E1 + 2 + 3 finally improved the luciferase activity, hinting that the function of fragment E1 + 2 was neutralized by fragment E3 (Figure 4j). Moreover, after co-transfected exon fragments with overexpression vector or ASO of *SMARCD3-OT1* in DF-1 cell line, the dual-luciferase assays proved that *SMARCD3-OT1* could improve the effects of exon 3 and exon 1 + 2 + 3, and they inhibited the negative effect of exon 1 + 2 (Figure 4k,l). Overall, the full-length of *SMARCD3-OT1* ultimately showed a positive regulation toward *SMARCD3X4*, suggesting that *SMARCD3-OT1* may recruit transcriptional factors to the promoter of *SMARCD3X4*, thus promoting the transcriptional activity of *SMARCD3X4*.

To investigate the potential transcriptional factor, we used the hTFtarget database [20] to screen for the transcriptional factors that could combine with the 2,000 bp upstream of *SMARCD3X4* (Appendix A). Then, the RIPseq website [21] was used to evaluate the potentialities of the combination between *SMARCD3-OT1* and these transcriptional factors. The results showed that the transcriptional factor SP2 had the potential to combine with the *SMARCD3-OT1* and 2,000 bp upstream of *SMARCD3X4* (Appendix A). These results indicate that SP2 may be the regulator that participates in the mechanism between *SMARCD3-OT1* and *SMARCD3X4*.

### 2.5. SMARCD3X4 Positively Regulates Myoblasts Proliferation

After 48 h transfection, the fold-changes in mRNA and protein levels were measured by qRT-PCR and Western blot assays (Appendix A). Being similar to *SMARCD3-OT1*, overexpression of *SMARCD3X4* significantly inhibited the expression level of *CDKN1A* and improved the expression of *CDKN1B*, *CCNA1*, *CCNE1*, and *CCND1*. In addition, the knockdown of *SMARCD3X4* had an inverse influence, revealing its positive influence on cell proliferation (Figure 5a,b). The cell cycle phase assay was performed to investigate the function mentioned above. After overexpression of *SMARCD3X4*, the number of cells in the S phase increased remarkably. Meanwhile, after the knockdown of *SMARCD3X4*, the number of cells in the S phase decreased, and the number of cells in the G2M phase showed an increase. These data demonstrated that *SMARCD3X4* improved cell proliferation by stopping cells from entering the G2M phase (Figure 5c,d). In addition, we performed the EdU assay to verify this conclusion. After 48 h transfection, the number of proliferating cells significantly increased with the overexpression of *SMARCD3X4* and decreased with the inhibition of *SMARCD3X4* (Figure 5e–h). Otherwise, the CCK-8 assay showed that *SMARCD3X4* promoted cell proliferation after 48 h transfection (Figure 5i,j). These data reveal that *SMARCD3X4* positively regulates myoblast proliferation after 48 h transfection.

### 2.6. SMARCD3X4 Positively Regulates Myoblasts Differentiation and Accelerates Myofiber Transformation

With respect to cell differentiation, qRT-PCR was used to detect the expression of differentiation-related genes with overexpression or inhibition of *SMARCD3X4* in CPMs after 3 days of differentiation. From the data, we observed that overexpression of *SMARCD3X4* could improve the expression of *MYOD*, *MYOG*, *MyHC*, and *MYF5*, and inhibition of *SMARCD3X4* repressed the expression level of these genes (Figure 6a,b). Similarly, at the protein level, *SMARCD3X4* expression significantly improved the expression of differentiation marker genes, promoting the differentiation of myoblasts through the *MYOD* pathway in differentiated myoblasts (Figure 6c,d). Meanwhile, the immunofluorescence staining assay illustrated that the cells overexpressed with *SMARCD3X4* had bigger myofibers, while the cell inhibited with *SMARCD3X4* showed smaller myofibers. These results demonstrated that *SMARCD3X4* facilitated the formation and maturity of the myofiber (Figure 6e–h). These cell experiments reveal that *SMARCD3X4* promotes myoblast differentiation after 3 days of differentiation, indicating that *SMARCD3X4* plays different roles in myoblast proliferation and differentiation stages. This result is similar to that of *SMARCD3-OT1*.

Otherwise, the qRT-PCR results also suggested that *SMARCD3X4* facilitated myofibers to transfer from slow-twitch myofibers to fast-twitch myofibers. Overexpression of *SMARCD3X4* increased the expression of fast-twitch myofiber-related genes and decreased the expression of slow-twitch myofiber-related genes. By contrast, the knockdown of *SMARCD3X4* showed a reverse influence (Figure 6i,j). 

### 2.7. SMARCD3-OT1 Regulates CDKN1A Pathway and MYOD Pathway In Vivo

The gastrocnemius muscles injected with overexpression lentivirus or modified ASO fragments of *SMARCD3-OT1* were detached and used for further experiments. The samples were detected by qRT-PCR; then the samples that were successfully overexpressed, or knockdown of *SMARCD3-OT1*, were collected and prepared for subsequent experiments. 

Seven-day-old chickens were induced (LV-*SMARCD3-OT1*) or knockdown (LV-NC) *SMARCD3-OT1* by lentivirus and specially modified ASO in order to investigate the role of *SMARCD3-OT1*. qRT-PCR was performed to detect the effect of lentivirus injection. As can be seen from Figure 7a,b, the fold changes of LV-*SMARCD3-OT1* and ASO are 28.7 and 0.3, respectively. qRT-PCR and Western blot were used to detect the expression of genes related to the *CDKN1A* pathway and *MYOD* pathway in vivo. The data showed that overexpression of *SMARCD3-OT1* downregulated the expression of *CDKN1A* and upregulated the expression of the downstream genes of *CDKN1A* in vivo, and the knockdown of *SMARCD3-OT1* had different results. Together, these data indicated that *SMARCD3-OT1* still regulated the *CDKN1A* pathway in vivo (Figure 7c,d). In addition, the qRT-PCR and Western blot results suggested that the myogenesis maker genes expressions increased with the expression of *SMARCD3-OT1* in mRNA and protein levels, indicating that *SMARCD3-OT1* still regulated the *MYOD* pathway in vivo (Figure 7e–h).

### 2.8. SMARCD3-OT1 Positively Regulates Muscle Development and Improves Muscle Hypertrophy and Myofibers Transformation

To further investigate the function of *SMARCD3-OT1* in myogenesis, the mass of the muscle injected with overexpressed lentivirus or modified ASO was measured. The data showed that the overexpression of *SMARCD3-OT1* led to an increased mass of leg muscle (Figure 8a). On the contrary, the knockdown of *SMARCD3-OT1* resulted in a decrease in leg muscle (Figure 8b). Meanwhile, the muscle was sliced up and used for hematoxylin and eosin (H&E) staining. The results revealed that muscle induced with *SMARCD3-OT1* had bigger myofibers, and the proportion of myofibers with a larger diameter was increased. By contrast, the muscle injected with modified ASO had smaller myofibers, and the proportion of myofiber with a smaller diameter was increased (Figure 8c–e). These data demonstrated that *SMARCD3-OT1* induced muscle hypertrophy and improved muscle development.

Immunohistochemical assays proved that *SMARCD3-OT1* enhanced the expression level of MYH1, a fast-twitch myofiber protein. The proportion of myofibers stained with MYH1 increased with the overexpression of *SMARCD3-OT1* and decreased with the inhibition of *SMARCD3-OT1* (Figure 8f,h,i). Conversely, the overexpression of *SMARCD3-OT1* decreased the proportion of myofibers stained with MYH7, a slow-twitch myofiber protein, and the inhibition of *SMARCD3-OT1* increased this proportion (Figure 8g,j,k). These data demonstrated the inhibitory effect of *SMARCD3-OT1* on MYH7. LDH is one of the important enzymes in anaerobic glycolysis and gluconeogenesis, which can reflect the level of anaerobic glycolytic capacity of skeletal muscle. The activity of LDH reflects the process of fast-twitch fiber transformation. The data showed that the activity of lactate dehydrogenase (LDH) was increased with the *SMARCD3-OT1* induced and suppressed with the knockdown of *SMARCD3-OT1*. These results indicated that *SMARCD3-OT1* facilitated the capacity of anaerobic glycolysis in skeletal muscle and promoted fast-twitch myofiber transformation (Figure 8l,m). Otherwise, Western blot was used to qualify the MYH1 and MYH7 protein content, which also revealed the positive influence of *SMARCD3-OT1* on the process of fast myofiber transformation (Figure 8n,o). The qRT-PCR results of myofiber transformation-related genes also proved this conclusion (Figure 8p,q).

## 3. Discussion

LncRNA plays an important role in various biological growth processes, affecting life processes, including bone development, muscle growth, fat deposition, disease resistance, tumorigenesis [22,23,24,25,26,27,28,29], and so on. Research about the regulatory mechanism of lncRNAs mainly focused on the competing endogenous RNAs way, in which lncRNA competitively bound the target gene of miRNA to reduce the effect of miRNA on target mRNA and positively regulated the expression of target genes [30,31,32,33]. However, reports about the relationship between lncRNAs and their adjacent functional genes are relatively rare.

*SMARCD3X4* is one of the transcripts of the *SMARCD3* gene, and the protein coded by *SMARCD3* is one of the components of the SWI/SNF complex, which is an evolutionarily conserved multi-subunit chromatin-remodeling complex, using the energy of ATP hydrolysis to mobilize nucleosomes and remodel chromatin [34]. The SWI/SNF complex can regulate transcription of target genes, regulating cell proliferation and differentiation [35,36,37].

In this study, we proved that *SMARCD3-OT1* can positively regulate the expression of *SMARCD3X4*, and the fragment E3 (965–1542) of *SMARCD3*-*OT1* is the main positive regulator of *SMARCD3X4*. The overexpression of *SMARCD3-OT1* significantly improves *SMARCD3X4* expression on the mRNA and protein level, which indicates that *SMARCD3-OT1* may influence mRNA transcription or translation of *SMARCD3X4* to some extent. We found that the transcriptional factor SP2 is the potential to combine with both lncRNA *SMARCD3-OT1* and 2000 bp upstream of SMARCD3X4 by using the target database and RIPseq website; thus, we speculated that SP2 may be the regulator that participates in the mechanism between *SMARCD3X4* and lncRNA *SMARCD3-OT1*.

Cell proliferation and differentiation are two different stages of cell processes. However, some studies report that they can be improved by the same genes. Circ-*RILPL1* can promote myoblast proliferation and differentiation via binding miR-145 and activating *IGF1R*/*PI3K*/*AKT* pathway [38]. Moreover, circ-*DAB1* is found to promote cell proliferation and osteogenic differentiation of human bone marrow stem cells via the *RBPJ*/*DAB1* axis [39]. In the cell experiments, we proved that *SMARCD3-OT1* and *SMARCD3X4* can promote myoblast proliferation in the proliferation stage (48 h after cells achieve 80–90%) and promote myoblast differentiation in the differentiation stage (3 days after differentiation induction). *SMARCD3-OT1* has different functions on myoblast proliferation and differentiation. A hypothesis is that SMARCD3-OT1 regulates myoblast proliferation and differentiation depending on its intracellular concentration. SMARCD3-OT1 expression in myoblast proliferation and differentiation results in a similar expression of SMARCD3X4 in these two stages. As a component of the SWI/SNF complex, SMARCD3X4 has the potential to regulate the SWI/SNF complex. The SWI/SNF complex can change the construction of chromatin, thus regulating the activation and transcription of functional genes and controlling the process of cell proliferation and differentiation. Therefore, we speculate that SMARCD3-OT1 may influence the SWI/SNF complex through SMARCD3X4, playing different roles in cell proliferation and differentiation.

Although several lncRNAs have been verified as regulatory factors in cell proliferation and differentiation, few of their functions about muscle hypertrophy and myofiber remodeling have been elaborated [40]. In this study, we proved that *SMARCD3-OT1* and *SMARCD3X4* can promote the expression of fast-twitch fiber-related genes and inhibit the expression of slow-twitch fiber-related genes in vitro. Moreover, we demonstrated that they could improve muscle hypertrophy and fast-twitch fiber formation in vivo.

In summary, our research reveals a novel lncRNA that can positively regulate the functional gene *SMARCD3X4*, facilitating myoblast proliferation and differentiated myoblast differentiation, improving myotube formation and promoting muscle hypertrophy and myofiber transformation (Figure 9). These findings, which offer us new insight into lncRNA and gene *SMARCD3*, contribute to the development of myogenesis research.

## 4. Materials and Methods

### 4.1. Ethics Statement

The animals used in the research were all fed and slaughtered under the guidance of the Institutional Animal Care and Use Committee at South China Agricultural University (approval ID: SCAU#2020C010, approval date: 1 June 2020). All experiments in the study were conducted under the supervision of South China Agricultural University, following international animal welfare standards. 

### 4.2. Cell Culture and Transfection 

DF-1 cell line was cultured in a growth medium consisting of Dulbecco’s Modified Eagle Medium (DMEM, Gibco, Carlsbad, CA, USA), supplemented with 10% fetal bovine serum (FBS, Gibco, Carlsbad, CA, USA) and 0.2% penicillin and streptomycin (Invitrogen, Carlsbad, CA, USA) [41]. Chicken primary myoblasts (CPMs) were isolated from leg muscle detached from E11 chickens. The myoblasts isolated from leg muscle were first cultured in a 37 °C incubator for 40 min. During this time, other cells would adhere to the petri dish, and the supernatant would be transferred to another petri dish. After three operations, the pure myoblasts could be obtained [42]. The myoblasts were cultured with Roswell Park Memorial Institute 1640 medium (RPMI 1640, Gibco, Carlsbad, CA, USA) supplemented with 20% FBS and 0.2% penicillin and streptomycin [43]. For inducing the differentiation of cells, the cell culture medium was replaced with a differentiated medium, RPMI 1640 medium with 2% horse serum and 0.2% penicillin and streptomycin, after which cells achieved 80–90% cell confluence. All cells were incubated by a temperature incubator with 37 °C and 5% CO_2_.

All transfection operations were performed with Lipofectamine 3000 reagent (Invitrogen, Carlsbad, CA, USA) and were operated under the guideline of the manufacturer’s protocol.

### 4.3. RNA Extraction, Complementary DNA (cDNA) Synthesis, and qRT-PCR

RNA of cells and tissues was extracted by TRIzol reagent (TaKaRa, Kusatsu, Japan) following the manufacturer’s protocol. Paris Kit (Ambion, Life Technologies, Austin, TX, USA) was used in nuclear and cytoplasmic RNA fractionation experiments with the guide of the manufacturer’s protocol. cDNA was synthesized with the PrimeScript RT Reagent Kit with Genomic DNA (gDNA) Eraser (perfect real-time) (TaKaRa, Kusatsu, Japan). The cDNA used for RACE was synthesized with the SMARTer RACE cDNA Amplification Kit (Clontech, Kusatsu, Japan). The qRT-PCR experiments were performed by using the iTaq Universal SYBR Green Supermix Kit (Bio-Rad, Hercules, CA, USA) and were analyzed by Bio-Rad Real-Time Detection Machine (Bio-Rad, Hercules, CA, USA). The chicken GAPDH gene was used as an internal control. Comparative ^2−^^ΔΔCT^ method was used to analyze the data from qRT-PCR [44]. Primers used in qRT-PCR are listed in Appendix A.

CPMs were transfected with the overexpression vector and ASO fragment of *SMARCD3-OT1* or *SMARCD3X4* when the cells achieved 80–90% confluence. After 48 h, the CPMs were collected and used for the detection of cell-cycle related genes, including *CDKN1A*, *CDKN1B*, *CCNA1*, *CCND1*, and *CCNE1*.

After 48 h transfection, CPMs were induced to differentiation by differentiated medium for 3 days. The RNA and protein of these cells were extracted and used for the detection of differentiation marker genes, including *MYOD*, *MYOG*, *MYHC*, and *MYF5*.

### 4.4. 5′ and 3′ RACE

The full length of *SMARCD3-OT1* was synthesized by using the SMARTer RACE cDNA Amplification Kit following the manufacturer’s protocol. The primer pairs used in RACE are listed in Appendix A.

### 4.5. Plasmid Construction and RNA Oligonucleotides

For mutation of the pEGFP-N1 vector (Promega, Madison, WI, USA), the full length of the pEGFP-N1 vector was amplified with a mutation in the start codon (ATGGTG to ATTGTT) [45]. The mutational ORFs of *SMARCD3-OT1* (in which the termination codon TGA was mutated to TGG) were amplified and cloned into the mutational pEGFP-N1 vector (pEGFP mut).

For overexpression vector construction, the full lengths of the *SMARCD3-OT1* sequence and *SMARCD3X4* coding sequence (NCBI: XM_040691998.1) were cloned and inserted into the pcDNA-3.1 vector (Promega, Madison, WI, USA) by utilizing the HindIII and XhoI restriction sites.

For dual-luciferase reporter vectors construction, the *SMARCD3-OT1 exon 1* (*SMARCD3-OT1*-*E1*, 1 bp–768 bp), *SMARCD3-OT1 exon 2* (*SMARCD3-OT1-E2*, 769 bp–964 bp), *SMARCD3-OT1 exon 3* (*SMARCD3-OT1-E3*, 965 bp–1542 bp), *SMARCD3-OT1 exon 1 + 2* (*SMARCD3-OT1-E1 + 2*, 1 bp–964 bp), and *SMARCD3-OT* (*SMARCD3-OT1-E1 + 2 + 3*, 1 bp–1542 bp), were cloned into the PGL3-promoter vector (Promega, Madison, WI, USA) by using the HindIII and NcoI restriction sites.

For the specific knockdown of *SMARCD3-OT1* and *SMARCD3X4*, the ASO were designed and produced by Guangzhou RiboBio (Guangzhou, China), respectively, for *SMARCD3-OT1* and *SMARCD3X4* that mainly existed in the cell nucleus.

The primers pairs and oligonucleotide sequences utilized in plasmid construction and interference are offered in Appendix A.

### 4.6. Western Blot Analysis

The Western blot assays were performed as previously reported [46]. The antibodies and their dilutions utilized in the Western blots are listed as follows: rabbit anti-SMARCD1/3 Ab (DF10125; Affinity Biosciences, Changzhou, China; 1:1000), rabbit anti-GAPDH (AP0063; Bioworld Technology, Bloomington, MN, USA; 1:10,000), mouse anti-MyHC (B103; DHSB, USA; 0.5 µg/mL), rabbit anti-MyoD1 (P10085; Bioss, Beijing, China; 1:500), rabbit anti-MYF5 (913349; Bioss, Beijing, China; 1:500), rabbit anti-Myogenin (P15173; Bioss, Biejing, China; 1:500), GFP-tag monoclonal antibody (AP0675M; Bioworld Technology, Bloomington, MN, USA; 1:2000), and beta-actin polyclonal antibody (Ap0060; Bioworld Technology, Bloomington, MN, USA; 1:10,000). Goat Anti-rabbit IgG-HRP (BA1054; Boster, Wuhan, China; 1:10,000) and Peroxidase-goat Anti-mouse IgG (BA1051; Boster, Wuhan, China; 1:10,000) were utilized as secondary antibodies.

### 4.7. Immunofluorescence

Cells were cultured and seeded into a 12-well plate and were transfected for 48 h. After 48 h transfection, the CPMs were induced to differentiation for 3 days. Then, the cells were fixed by 4% formaldehyde for 30 min and were washed by phosphate-buffered saline (PBS, Gibco, Waltham, MA, USA) for 15 min. Next, the cells’ cytomembranes were destroyed by 0.1% Triton X-100 diluted by PBS for 20 min and were blocked by goat serum for 20–30 min. After that, cells were incubated with mouse anti-MyHC (B103; DHSB, Iowa City, IA, USA; 0.5 µg/mL) overnight at 4 °C, and were incubated with Fluorescein (FITC)-conjugated AffiniPure Goat Anti-Mouse IgG (H + L) (BS50950; Bioworld, USA; 1:50) at room temperature for 1 h. The cell nucleus was stained by DAPI (Solarbio, Beijing, China) for 10 min. A TE2000-U fluorescence microscope (Nikon, Tokyo, Japan) was utilized to capture the image of cells, and the images were analyzed by ImageJ software (National Institutes of Health, Bethesda, MD, USA).

### 4.8. Flow Cytometry, EdU, and CCK-8 Assays

For the analysis of cell proliferation stages, cells were cultured and seeded into a 12-well culture plate and transfected for 48 h. After pre-cooling PBS washing, cells were fixed in 70% ethanol and stored at −20 °C overnight. Then, a Cell Cycle Analysis Kit (Thermo Fisher Scientific, MA, USA) was used in flow cytometry analysis of the cell cycle. A BD Accuri C6 flow cytometer (BD Biosciences, San Jose, CA, USA) was utilized for the analysis of myoblasts. The FlowJo software (7.6, Tree Star, Ashland, OR, USA) was used for data processing.

For the EdU assay, myoblasts were seeded into a 12-well culture plate and transfected when the cells density reached 70–80%. After 48 h transfection, cells were fixed by 4% formaldehyde for 30 min and stained by using EdU Apollo In vitro Imaging Kit (C10310, RiboBio, Guangzhou, China). A fluorescence microscope was utilized to acquire images, and ImageJ software was used for the analysis of data. 

For the CCK-8 assay, primary myoblasts were seeded into 96-well culture plates and transfected. Then, the cell growth condition was monitored at 12, 24, 36, and 48 h by using the TransDetect CCK Kit (TransGen Biotech, Beijing, China) following the instruction book. The absorption spectra at 450 nm were detected by using an iMark microplate absorbance reader (Bio-Rad, Hercules, CA, USA). All data were acquired from six independent repeats.

### 4.9. Lentivirus Assay

Thirty 7-day-old chickens from Yu He Agriculture and Animal Husbandry Co., Ltd. were fed in the same room with free food as well as water and were randomly divided into two groups (*n* = 15): (1) *LV*-*SMARCD3-OT1* and LV-NC, and (2) ASO-*SMARCD3-OT1* and ASO-NC. Chickens were injected with lentivirus (3 × 10^6^ titers) or modified ASOs (40 nmol) by intramuscular injection on days 7 and 14. The overexpressed lentivirus and ASO-*SMARCD3-OT1* were injected into the chickens’ left gastrocnemius muscle, and the reagents of control groups were injected into the right gastrocnemius muscle. The chickens were euthanized at 21 days old, and the gastrocnemius muscles were detached and stored at −80 °C.

### 4.10. H&E Staining and Immunohistochemistry

For H&E staining, gastrocnemius muscles were fixed in 4% paraformaldehyde overnight and sent to Servicebio Co., Ltd. (Wuhan, China) for slicing up and H&E staining.

For immunohistochemistry, the sections of gastrocnemius muscle tissues were stained by using an SP-POD kit (SP0041, Solarbio, Beijing, China). The primary antibodies were anti-MYH1 (GTX17458; Genetex, Irvine, CA, USA; 1:400) and anti-MYH7 (S58; DHSB, Iowa City, IA, USA; 1:100).

### 4.11. Enzyme Activities Analysis

The gastrocnemius muscles induced or knockdown of *SMARCD3-OT1* were collected and cut into small pieces, which were subsequently weighed and ground at 4 °C. Then, the ground samples were centrifuged, and the supernatant was collected for the subsequent detection of the activity of LDH. The LDH Activity Detection Kit (BC0685, Solarbio, Beijing, China) was used to detect the activity of LDH. The data were acquired by a fluorescence/multi-detection microplate reader (BioTek, Winooski, VT, USA) at 450 nm, and the activity of LDH was calculated based on the weight of the sample.

### 4.12. Dual-Luciferase Reporter Assay

The PGL3-promoter inserted with the different parts of *SMARCD3-OT1* was transfected into a DF-1 cell line in a 96-well culture plate. After 48 h transfection, a Dual-Glo Luciferase Assay System Kit (Promega, Madison, WI, USA) was used to detect the firefly and Renilla luciferase activities following the manufacturer’s protocol. The data were acquired by a fluorescence/ multi-detection microplate reader (BioTek, Winooski, VT, USA). All data were acquired from eight independent repeats.

### 4.13. Statistical Analysis

Every assay was repeated at least three times. The data from every experiment were presented by mean ± S.E.M, and the statistical significance of differences between different groups was tested by independent or paired *t* tests. Independent *t* tests were used in cell experiments, and paired *t* tests were used in animal experiments. * *p* < 0.05 was considered as significant, and ** *p* < 0.01 was considered as highly significant.

## Figures and Tables

**Figure 1 ijms-23-04510-f001:**
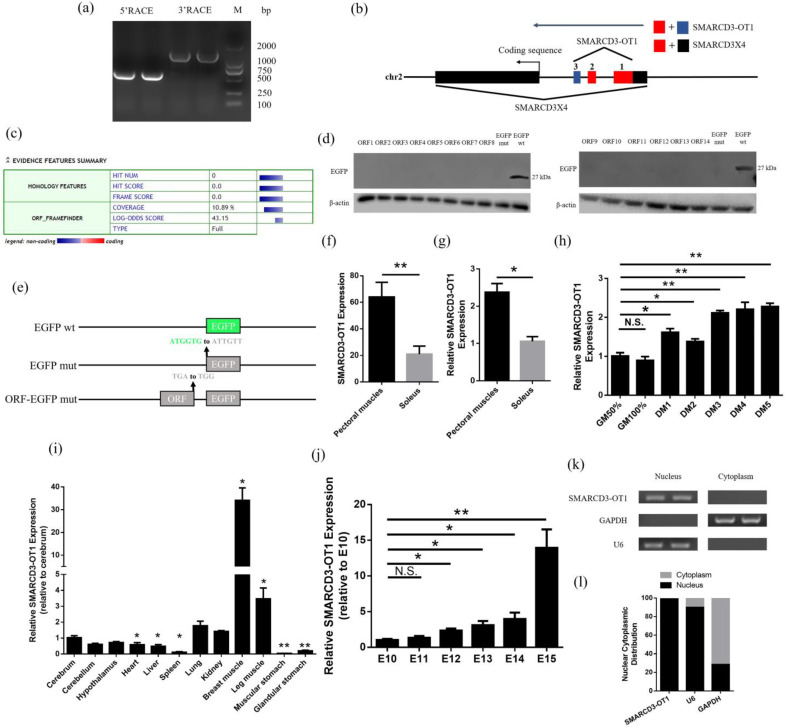
Identification of lncRNA *SMARCD3-OT1* and the expression pattern of *SMARCD3-OT1*. (**a**) Results of *SMARCD3-OT1* 5′ RACE and 3′RACE. (**b**) Schematic image of the locations for *SMARCD3-OT1* (red and blue) and *SMARCD3X4* (red and black). Arrows represent the direction of transcription. (**c**) Prediction of the *SMARCD3-OT1* protein-coding ability. (**d**) Western blotting with anti-EGFP. (**e**) Diagram of the EGFP fusion construct vectors used for transfection. The initiation codon ATGGTG of the EGFP (EGFP wt) gene is mutated to ATTGTT (EGFP mut), and the termination codon TGA of ORFs is mutated to TGG. (**f**,**g**) RNA sequencing result of *SMARCD3-OT1* and further qRT-PCR verification. (**h**) Expression levels of *SMARCD3-OT1* in CPMs cultured in growth medium at 50% and 100% cell confluency (50%GM and 100%GM) and differentiation medium from 1 to 5 days (DM1 to DM5) (*n* = 6). (**i**) Tissue expression profiles of *SMARCD3-OT1* in 7-week-old chickens (*n* = 6). (**j**) Expression levels of *SMARCD3-OT1* from E10 to E15 (*n* = 6). (**k**,**l**) Distribution of *SMARCD3-OT1* in the cytoplasm and nuclei of CPMs was determined by PCR and qRT-PCR. Data are presented as mean ± SEM. Statistical significance of differences between means was assessed using an independent sample *t* test (* *p* < 0.05; ** *p* < 0.01; N.S., no significant difference).

**Figure 2 ijms-23-04510-f002:**
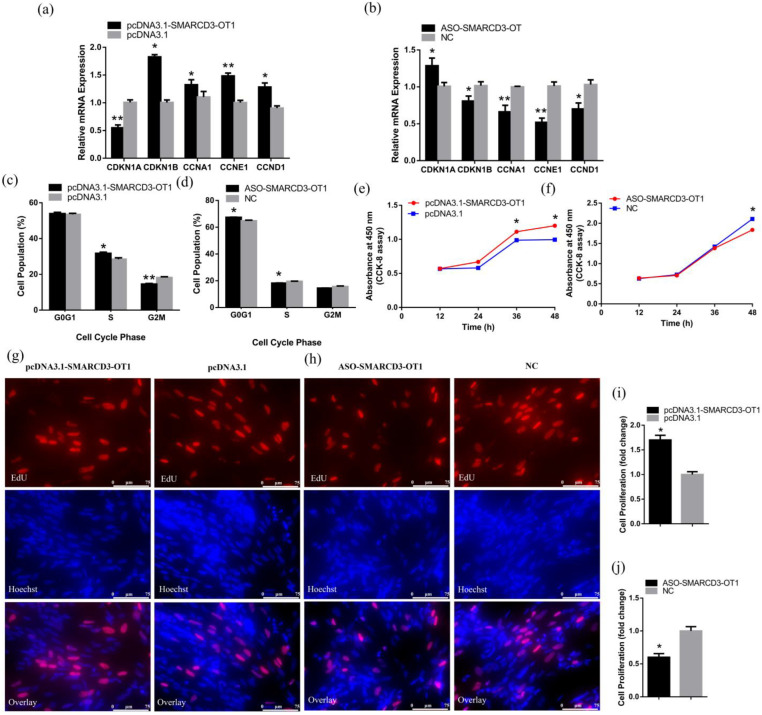
*SMARCD3-OT1* promotes the proliferation of myoblasts. (**a**,**b**) Relative mRNA levels of several cell cycle genes after overexpression or knockdown of *SMARCD3-OT1* (*n* = 6). (**c**,**d**) Cell cycle analysis of CPMs at 48 h after overexpression or knockdown of *SMARCD3-OT1* (*n* = 4). (**e**,**f**) CCK-8 assays were performed in CPMs with *SMARCD3-OT1* overexpression or knockdown (*n* = 6). (**g**,**h**) After 48 h transfection of overexpression vector or ASO of *SMARCD3-OT1*, CPMs were stained by EdU and Hoechst, and the images were captured by a fluorescence microscope (*n* = 6). (**i**,**j**) Proliferation rate of CPMs with *SMARCD3-OT1* overexpression or knockdown (*n* = 6). All proliferation-related experiments are performed in CPMs during the cells’ proliferation stage without myoblast fusion. Data are expressed as a fold change relative to the control. Results are shown as mean ± SEM. Statistical significances of differences between means were assessed using an independent sample *t* test. * *p* < 0.05, ** *p* < 0.01.

**Figure 3 ijms-23-04510-f003:**
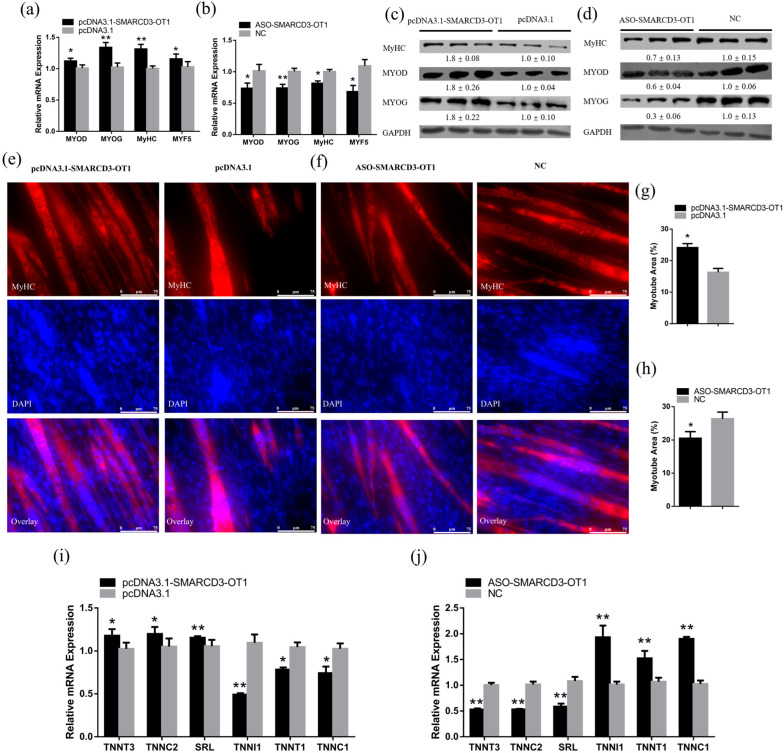
*SMARCD3-OT1* promotes the differentiation of myoblasts during the differentiation stage, facilitating myofiber transformation. (**a**,**b**) Relative mRNA levels of several cell differentiation marker genes after overexpression or knockdown of *SMARCD3-OT1* (*n* = 6). (**c**,**d**) Relative protein levels of several cell differentiation marker genes after overexpression or knockdown of *SMARCD3-OT1* (*n* = 3). The numbers shown below the bands were folds of band intensities relative to control. Band intensities were quantified by ImageJ and were normalized to GAPDH. (**e**,**f**) MyHC immunostaining of CPMs transduced with *SMARCD3-OT1* overexpression or knockdown. Cells were differentiated for 72 h after transfection. The nuclei were visualized with 4′,6-diamidino-2-phenylindole buffer (DAPI) (*n* = 6). (**g**,**h**) Myotube area (%) of CPMs transduced with *SMARCD3-OT1* overexpression or knockdown in immunofluorescence assay (*n* = 6). (**i**,**j**) Relative mRNA levels of several fast-twitch fiber- and slow-twitch fiber-related genes after overexpression or knockdown of *SMARCD3-OT1* (*n* = 6). All differentiation-related experiments are performed in CPMs after 3 days of differentiation induction using the differentiated medium. Data are expressed as a fold change relative to the control. Results are shown as mean ± SEM. Statistical significances of differences between means were assessed using an independent sample *t* test. * *p* < 0.05, ** *p* < 0.01.

**Figure 4 ijms-23-04510-f004:**
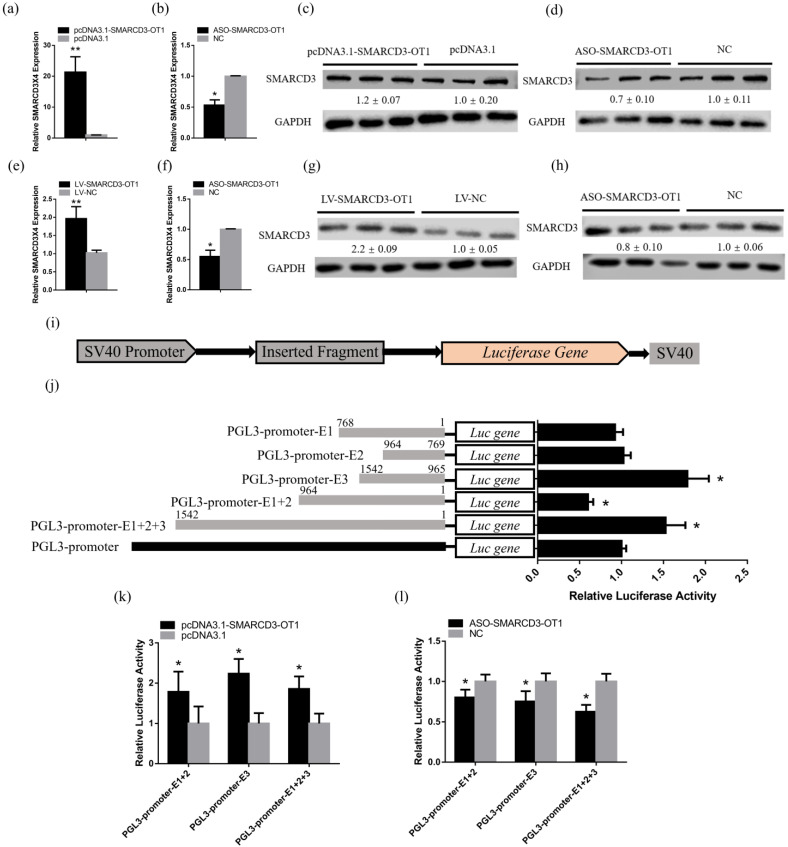
*SMARCD3-OT1* positively regulated *SMARCD3X4*. (**a**,**b**) mRNA expression levels of *SMARCD3X4* in CPMs with *SMARCD3-OT1* overexpression or knockdown (*n* = 6). (**c**,**d**) Protein expression levels of SMARCD3X4 in CPMs with *SMARCD3-OT1* overexpression or knockdown (*n* = 3). The numbers shown below the bands are folds of band intensities relative to control. Band intensities were quantified by ImageJ and normalized to GAPDH. (**e**,**f**) mRNA expression levels of *SMARCD3X4* in animals with *SMARCD3-OT1* overexpression or knockdown (*n* = 6). (**g**,**h**) Protein expression levels of *SMARCD3X4* in animals with *SMARCD3-OT1* overexpression or knockdown (*n* = 3). The numbers shown below the bands are folds of band intensities relative to control. Band intensities were quantified by ImageJ and normalized to GAPDH. (**i**) Sketch of the PGL3-promoter vector inserted with *SMARCD3-OT1* fragments. (**j**) After 48 h, the relative luciferase activities of DF-1 cell transfected with a different recombinant vector were detected (*n* = 8). (**k**,**l**) Relative luciferase activities of DF-1 cell co-transfected with different recombinant vectors and overexpression vector or ASO of *SMARCD3-OT1*. Data are presented as mean ± SEM. Statistical significance of differences between means was assessed using an independent sample *t* test. * *p* < 0.05, ** *p* < 0.01.

**Figure 5 ijms-23-04510-f005:**
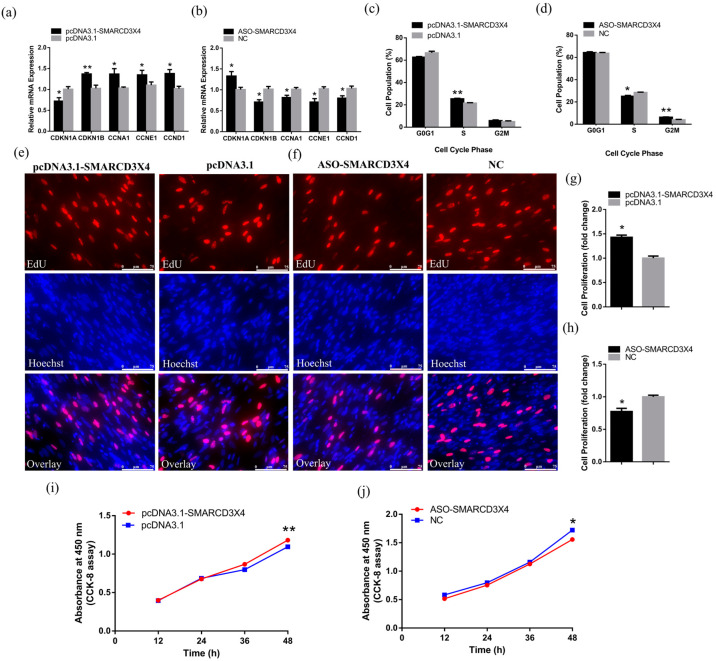
*SMARCD3X4* promotes cell proliferation. (**a**,**b**) Relative mRNA levels of several cell cycle genes after overexpression or knockdown of *SMARCD3X4* (*n* = 6). (**c**,**d**) Cell cycle analysis of CPMs after overexpression or knockdown of *SMARCD3X4* (*n* = 4). (**e**,**f**) EdU proliferation assays for CPMs with the overexpression or knockdown of *SMARCD3X4* (*n* = 6). (**g**,**h**) Proliferation rate of CPMs with *SMARCD3X4* overexpression or knockdown (*n* = 6). (**i**,**j**) CCK-8 assays were performed in CPMs with *SMARCD3X4* overexpression or knockdown (*n* = 6). All proliferation-related experiments are performed in CPMs during the cells’ proliferation stage without myoblast fusion. Data are expressed as a fold change relative to the control. Results are shown as mean ± SEM. Statistical significances of differences between means were assessed using an independent sample *t* test. * *p* < 0.05, ** *p* < 0.01.

**Figure 6 ijms-23-04510-f006:**
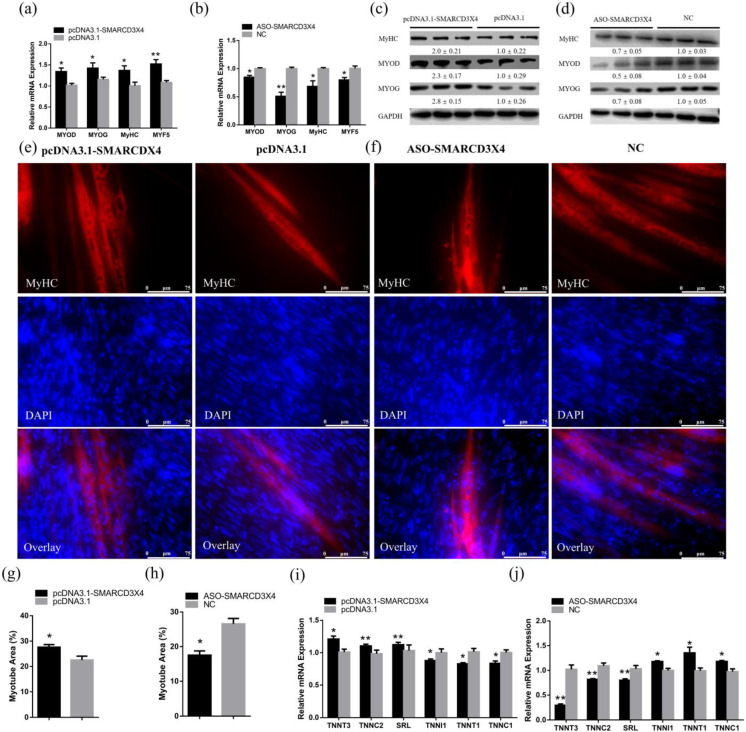
*SMARCD3X4* promotes cell differentiation through the *MYOD* pathway and accelerates myofiber transformation during myoblast differentiation. (**a**,**b**) Relative mRNA levels of several cell differentiation marker genes after overexpression or knockdown of *SMARCD3X4* (*n* = 6). (**c**,**d**) Relative protein levels of several cell differentiation marker genes after overexpression or knockdown of *SMARCD3X4* (*n* = 3). The numbers shown below the bands are folds of band intensities relative to control. Band intensities were quantified by ImageJ and normalized to GAPDH. (**e**,**f**) MyHC immunostaining of CPMs with *SMARCD3X4* overexpression or knockdown. (**g**,**h**) Myotube area (%) of CPMs transduced with *SMARCD3X4* overexpression or knockdown in immunofluorescence assay (*n* = 6). (**i**,**j**) Relative mRNA levels of several fast-twitch fiber- and slow-twitch fiber-related genes after overexpression or knockdown of *SMARCD3X4* (*n* = 6). All differentiation-related experiments are performed in CPMs after 3 days of differentiation induction using the differentiated medium. Data are expressed as a fold change relative to the control. Results are shown as mean ± SEM. Statistical significances of differences between means were assessed using an independent sample *t* test. * *p* < 0.05, ** *p* < 0.01.

**Figure 7 ijms-23-04510-f007:**
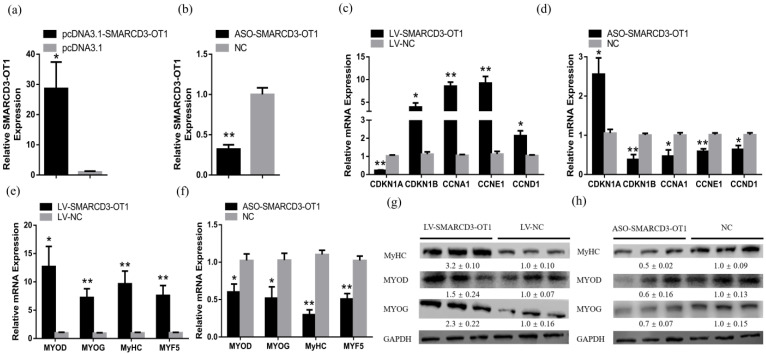
*SMARCD3-OT1* regulates the *CDKN1A* pathway and *MYOD* pathway in vivo. (**a**) Relative *SMARCD3-OT1* expression in gastrocnemius muscle after infection with *SMARCD3-OT1*-expressing lentivirus (LV-*SMARCD3-OT1*) or negative control (LV-NC) (*n* = 6). (**b**) Relative *SMARCD3-OT1* expression in gastrocnemius muscle after infection with modified ASO specific to *SMARCD3-OT1* (ASO-*SMARCD3-OT1*) or negative control (NC) (*n* = 6). (**c**,**d**) Relative mRNA levels of several cell cycle genes after overexpression or knockdown of *SMARCD3-OT1* in vivo (*n* = 6). (**e**,**f**) Relative mRNA levels of several myogenesis marker genes after overexpression or knockdown of *SMARCD3-OT1* in vivo (*n* = 6). (**g**,**h**) Relative protein levels of several myogenesis marker genes after overexpression or knockdown of *SMARCD3-OT1* in vivo (*n* = 3). The numbers shown below the bands are folds of band intensities relative to control. Band intensities were quantified by ImageJ and normalized to GAPDH. Data are expressed as a fold change relative to the control. Results are shown as mean ± SEM. Statistical significances of differences between means were assessed using a paired sample *t* test. * *p* < 0.05, ** *p* < 0.01.

**Figure 8 ijms-23-04510-f008:**
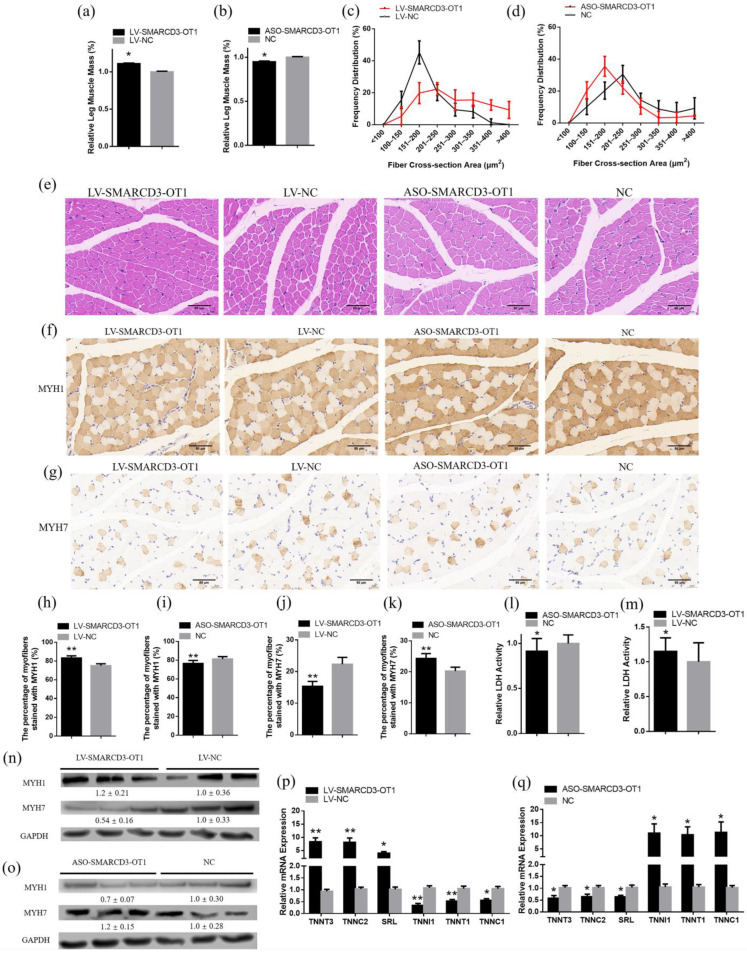
*SMARCD3-OT1* positively regulates muscle development and improves muscle hypertrophy and myofiber transformation. (**a**,**b**) Relative mass of gastrocnemius muscles after injection with overexpression or knockdown of *SMARCD3-OT1* in vivo (*n* = 15). (**c**–**e**) H&E staining and frequency distribution of fiber cross-section areas in gastrocnemius muscle with *SMARCD3-OT1* overexpression or knockdown (*n* = 6). (**f**,**h**,**i**) Immunohistochemistry of MYH1 and the percentage of myofibers stained with MYH1 protein with *SMARCD3-OT1* overexpression or knockdown (*n* = 6). (**g**,**j**,**k**) Immunohistochemistry of MYH7 and the percentage of myofibers stained with MYH7 protein with *SMARCD3-OT1* overexpression or knockdown (*n* = 6). (**l**,**m**) Relative enzymes activity of LDH in gastrocnemius muscle with *SMARCD3-OT1* overexpression or knockdown (*n* = 6). (**n**,**o**) Relative protein level of MYH1 and MYH7 in *SMARCD3-OT1*-induced or knockdown muscle (*n* = 3). The numbers shown below the bands are folds of band intensities relative to control. Band intensities were quantified by ImageJ and normalized to GAPDH. (**p**,**q**) Relative mRNA levels of several fast-twitch fiber- and slow-twitch fiber-related genes after overexpression or knockdown of *SMARCD3-OT1* in vivo (*n* = 6). Data are expressed as a fold change relative to the control. Results are shown as mean ± SEM. Statistical significances of differences between means were assessed using a paired sample *t* test. * *p* < 0.05, ** *p* < 0.01.

**Figure 9 ijms-23-04510-f009:**
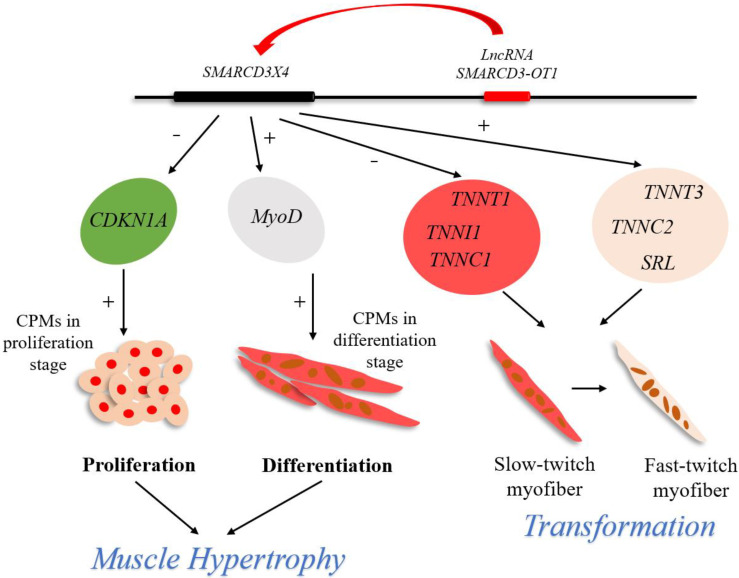
Model of lncRNA *SMARCD3-OT1* regulatory network for muscle development. Briefly, lncRNA *SMARCD3*-*OT1* upregulates the expression of *SMARCD3X4*, thus promoting the proliferation of myoblasts and the differentiation of the myoblasts after 3 days of differentiation induction. In addition, lncRNA *SMARCD3*-*OT1* improves muscle hypertrophy and fast-twitch fiber transformation in animals.

## Data Availability

The data presented in this study are available on request from the corresponding author.

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
