# Peer review of "LncRNA SMARCD3-OT1 Promotes Muscle Hypertrophy and Fast-Twitch Fiber Transformation via Enhancing SMARCD3X4 Expression"

_ijms, 2022, doi:10.3390/ijms23094510_

Round 1
Reviewer 1 Report
In the article by Zhang et al, the authors identify a lncRNA SMARCD3-OT1 which they find is highly expressed in muscle tissues and cells. Overexpression of SMARCD3-OT1 was surprisingly associated with both increased proliferation of myoblasts yet also improved differentiation into myotubes. The expression of SMARCD£-OT1 was tightly associated with SMARCD3X4, which they demonstrated to also improve proliferation of myoblasts and differentiation. The authors identify a region of SMARCD3-OT1 that promotes SMARCD3X4 activation. The authors suggest this pathway regulates the expression of cell cycle genes and promotes the expression of genes related to fast twitch fibres which they suggest is responsible to the increase in myotube size and diameter. The authors also use an in vivo model where they overexpress or inhibit SMARCD3-OT1 using lentivirus and results would indicate that overexpression promotes muscle hypertrophy. Further bioinformatics analysis presented in the discussion suggest that the transcription factor SP2 is recruited by SMARCd3-OT1 to promote the transcription of SMARCD3X4. Some of the work with the luciferase reporter and different fragments of SMARCD3-OT1 is quite elegant.
Major points:
There is some very interesting data presented in the article and the in vivo model is very convincing on the beneficial effects of SMARCD3-OT1, however some of the other data presented using the chicken primary myoblasts is contradictory. An improvement in myoblast proliferation and also differentiation is mutually exclusive and needs to be resolved. In order to differentiate and form myotubes, the myoblasts need to start undergoing fusion, this will not occur when they have increased proliferation. So it is very strange that something can promote proliferation of myoblasts can also promote differentiation. I believe this confusion may be due to the experimental design and exactly when the treatments of the myoblasts and myotubes occurred. The materials and methods needs to precisely describe exactly when and for how long the transfections were performed in both myoblasts and myotubes, and at exactly what stages the protein and mRNA extracts were extracted. This is critical as treatment of myoblasts and myotubes with the same chemical can have very different effects and at the moment based on information provided it is impossible to tell when these treatments occurred, for how long and at what stages samples were prepared.
Similarly for the expression of some of the markers of differentiation such as MyoD and MyoG e,g, MyoD is highly expressed in the early stages of differentiation and then decreases. These have very different expression profiles during the course of differentiation so when the samples were taken needs to be included.
The bioinformatics identifying SP2 as the potential link between SMARCD3-OT1 and SMARCD3X4 should be presented in more detail in the results section.
The schematic figure in the supplementary files should be improved based on comments above and included in main manuscript as it will aid readers.
In the text on a number of occasions including the abstract the authors state that SMARCD3-OT1 (or SMARCD3X3) “promotes proliferation and differentiation”, these statements need to be changed as again these are mutually exclusive.
Resolution of the myotube images is quite poor and good be significantly improved, please include scale bars also.
In statistics it says “differences between groups was tested by one- sample or paired t-tests”, I don’t understand how a paired t test could be used for the myoblasts or myotubes? Overall the materials and methods section needs a lot more specific detail on the procedures performed.
Minor points:
Article is littered with small grammatical errors that should be corrected during editing.
Author Response
Dear reviewer,
Thank you for your comments and suggestions. The revised manuscript was uploaded to the system, and the amendments are tracked in the revised version. The point-by-point responses were attached with this note. Please see the attachment.
Best wishes,
Jing Zhang

Reviewer 2 Report
The manuscript entitled “LncRNA SMARCD3-OT1 promotes myoblasts proliferation and differentiation via enhancing SMARCD3X4 expression in vitro and in vivo” is an interesting and novel paper by Zhang et al., which utilizes innovative technologies to investigate the mechanisms of the lncRNA SMARCD3-OT1 on myogenesis and myofiber transformation. This study by Zhang et al. identifies a new lncRNA SMARCD3-OT1, which is highly expressed in skeletal muscle. lncRNA SMARCD3-OT1 can upregulate the expression of SMARCDX4, thus promoting the myoblast proliferation and differentiation through CDKN1A and MYOD pathways, respectively. Moreover, lncRNA SMARCD3-OT1 can facilitate the slow-twitch fiber to transform into the fast-twitch fiber. The functional assays were well designed and strongly supported the main conclusion. The manuscript is well organized, and I would like to recommend this paper to be published in the International Journal of Molecular Sciences after the following points are addressed:
- The previous RNA-sequencing showed that lncRNA SMARCD3-OT1was highly expressed in chicken pectoral muscle. Why does the myoblast separate by leg muscle? Please explain this point.
- In figure 1b, it is hard to see the lncRNA SMARCD3-OT1located in the 5’ UTR of SMARCD3X4. Please point out the initiation codon of the SMARCD3X4.
- Were the qRT-PCR results of cell cycle-related genes and cell differentiation marker genes obtained from the different stages of myoblast? If so, please state this in your manuscript clearly.
- The quality of figure 3e (MYHC staining in cell transfect with pcDNA3.1) is poor, and there are a lot of red points in this figure. Please change a clearer one.
- The author performed a prediction using the hTFtarget database in the discussion. The sequence of the 2000 bp upstream of SMARCD3X4should be put into the supplementary table.
- The manuscript points out that cell purification is achieved by differential adhesion. Please list references to this method.
- The author should check the international unit in the whole manuscript. There should be a blank between the number and unit (for example, line 357).
- The author should check the whole manuscript and add a comma after the thousand digits.
Author Response

(The authors gave the same response as above.)

Round 2
Reviewer 1 Report
I don't believe the authors have addressed the fundamental issue that was raised in the initial review report i.e. that proliferation and differentiation of myoblasts are mutually exclusive and how a substance/condition can promote both proliferation and differentiation as initially claimed in the article has not been addressed. Although I do believe the current results are is due to the experimental design, this need to be teased out in relation to the design and results. Again this needs to be discussed in detail. In the revised manuscript the authors have just basically edited out any mention of proliferation from the text, although the results remain unchanged and include the figures on proliferation of myoblasts. Again, the resolution of the myotubes could be improved.
Author Response

(The authors gave the same response as above.)
